# Otic Capsule Dehiscences Simulating Other Inner Ear Diseases: Characterization, Clinical Profile, and Follow-Up—Is Ménière's Disease the Sole Cause of Vertigo and Fluctuating Hearing Loss?

Joan Lorente-Piera [1,*], Carlos Prieto-Matos [1], Raquel Manrique-Huarte [1], Octavio Garaycochea [1], Pablo Domínguez [2] and Manuel Manrique [1]

1 Otorhinolaryngology Department, University of Navarra Clinic, 31008 Pamplona, Navarra, Spain
2 Radiology Department, University of Navarra Clinic, 31008 Pamplona, Navarra, Spain
* Correspondence: jlorentep@unav.es; Tel.: +34-948-25-54-00 (ext. 4652)

**Abstract:** Introduction: We present a series of six cases whose clinical presentations exhibited audiovestibular manifestations of a third mobile window mechanism, bearing a reasonable resemblance to Ménière's disease and otosclerosis. The occurrence of these cases in such a short period has prompted a review of the underlying causes of its development. Understanding the pathophysiology of third mobile window syndrome and considering these entities in the differential diagnosis of conditions presenting with vertigo and hearing loss with slight air-bone gaps is essential for comprehending this group of pathologies. Materials and Methods: A descriptive retrospective cohort study of six cases diagnosed at a tertiary center. All of them went through auditive and vestibular examinations before and after a therapeutic strategy was performed. Results: Out of 84 cases of dehiscences described in our center during the period from 2014 to 2024, 78 belonged to superior semicircular canal dehiscence, while 6 were other otic capsule dehiscences. Among these six patients with a mean age of 47.17 years (range: 18–73), all had some form of otic capsule dehiscence with auditory and/or vestibular repercussions, measured through hearing and vestibular tests, with abnormalities in the results in five out of six patients. Two of them were diagnosed with Ménière's disease (MD). Another two had cochleo-vestibular hydrops without meeting the diagnostic criteria for MD. In two cases, the otic capsule dehiscence diagnosis resulted from an intraoperative complication due to a gusher phenomenon, while in one case, it was an accidental radiological finding. All responded well to the proposed treatment, whether medical or surgical, if needed. Conclusions: Otic capsule dehiscences are relatively new and unfamiliar entities that should be considered when faced with cases clinically suggestive of Ménière's disease, with discrepancies in complementary tests or a poor response to treatment. While high-sensitivity and specificity audiovestibular tests exist, completing the study with imaging, especially petrous bone CT scans, is necessary to locate and characterize the otic capsule defect responsible for the clinical presentation.

**Keywords:** dehiscence; otic capsule; third mobile window syndrome; Ménière disease; vertigo; hearing loss

## 1. Introduction

The first hypothesis and description of the third mobile window syndrome can be attributed to Tullio in the late 20th century, when he studied semicircular canal dehiscences in pigeons [1]. Hence, the phenomenon is known as Tullio's phenomenon, where clinical manifestations of vertigo and/or nystagmus are induced by acoustic changes. However, it was not until 1998 that Minor et al. [2] reported the documented consequences of the most studied dehiscence, that of the superior semicircular canal (SSC), from an audiovestibular perspective. Firstly, this dehiscence led to a conductive hearing loss and an increase in interaural symmetry or a reduction in threshold, detected by vestibular-evoked myogenic potentials (VEMP).

Otic capsule dehiscences may have an embryological origin, resulting from a disruption in the development of the otic capsule [3]. The involvement of the auditory canal and facial nerve in these malformations suggests that they may be congenital disorders rather than acquired ones related to a history of head trauma, which is one of the most common acquired etiologies, along with other factors such as increased intracranial pressure or tumors secondary to the inner ear [4].

Despite the Bárány Society consensus statement [5] that includes SSC dehiscence, perilymphatic fistula, or dilated vestibular aqueduct within the third-window phenomenon, since 2015, other entities with a similar clinical pattern have emerged [6–8] and must be considered in the differential diagnosis of Ménière's disease (MD). To attempt to explain why these discontinuities were causing symptoms, in 2008, Merchant and Rosowski [9] proposed a general hypothesis to explain the underlying mechanism of hearing loss associated with these anomalies. The typical propagation of sound occurs through the oval and round windows, acting as fluid interfaces between the air in the middle ear and the perilymphatic fluid spaces of the inner ear. Various conditions can enlarge existing bony channels or create additional defects in the osseous labyrinth, leading to the formation of hydrodynamic third windows [10]. Ultimately, given that temporal bone CT imaging is not notably complex, it may be prudent to explore the potential of otic capsule dehiscence in individuals experiencing vertigo symptoms alongside a minor discrepancy between air and bone conduction.

## 2. Materials and Methods

### 2.1. Study Design

A retrospective and descriptive cohort study with six cases spanning from 2018 to 2023.

### 2.2. Inclusion Criteria

Patients with vestibular and/or auditory alterations, regardless of the involvement of other structures in the otolaryngological area. Follow-up and recording of results before and after treatment based on the etiology and specific characteristics of the patient. Informed consents were obtained from the patients, and they agreed to participate in the study following the 1975 Helsinki Declaration.

### 2.3. Complementary Tests

All included patients underwent a physical examination, including otoscopy, and an otoneurological examination using a videonystagmography system (VideoFrenzel, Interacoustics VF505m, Middelfart, Denmark). For the presented cases, audiological tests (AC40, Interacustics, Middelfart, Denmark) were conducted to assess possible hearing loss. Findings are reported in terms of pure-tone thresholds from 0.25 to 6 kHz expressed in decibels of hearing level (dB HL). The degree of hearing loss was classified according to the criteria of the Bureau International d'Audiophonologie (BIAP). With speech audiometry, results are presented in the sound field at various presentation levels to determine the individual speech intensity function curve, including 65 dB SPL. Vestibular dysfunction was documented through recording vestibulo-ocular reflex (vHIT GN Otometrics, Taastrup, Denmark) and vestibular-evoked myogenic potentials (VEMPs). In both cervical (cVEMP) and ocular (oVEMP) tests, normal vestibular function is defined as the presence of a vestibular evoked myogenic potential in both ears. It will be analyzed by the interaural asymmetry ratio (IAAR (%)) for air-conducted stimulation at 0.5 kHz, 1 kHz, and 4 kHz. The intensity of the acoustic stimulus used will be 97 dB normalized HL. A Blackman envelope was configured (rise/fall time: 2 ms; plateau time: 0 ms). One hundred averages were presented at a rate of 5.1/s. The cVEMP is recorded with the patients seated in an upright position. The signals obtained were rectified by the contraction value of the ECM (sternocleidomastoid) muscle. The oVEMP is recorded with a patient sitting upright with their head in front and being instructed to look at a fixed point on the wall with an upwards inclination of 35°.

All cases also underwent computed tomography (CT) with 0.6 mm thick slices, along with three-dimensional segmentation of complete ear structures. In these cases, an axial plane is preferable for segmentation of these structures by setting a grayscale threshold to avoid capturing unwanted structures. In patients with suspected endolymphatic hydrops justifying their audiovestibular clinical symptoms, magnetic resonance imaging (3T) was performed with the T2-FLAIR sequence 4 hours after intravenous administration of gadolinium to grade cochlear and vestibular endolymphatic hydrops (EH).

*2.4. Treatment*

Pharmacological approaches were tailored to the specific pathology being treated and were in accordance with the latest clinical guidelines for each condition. Oral corticosteroid therapy at mg/kg was used to address auditory fluctuations in patients presenting with them. For long-term treatment, a diuretic like acetazolamide was employed, with the dosage depending on the patient's characteristics, to alleviate audiovestibular symptoms. Patients with severe-profound hearing loss were treated with a perimodiolar cochlear implant following standard procedures.

*2.5. Postoperative Follow-Up*

Postoperative implant follow-up includes medical appointments at one week, three weeks (implant activation), three months, six months, and one year. Continuous monitoring of the otologic cause is maintained. The study encompasses medical history, pre- and post-implant audiometric analysis, surgery details, and potential complications, along with postoperative free-field and speech audiometry.

**3. Results**

From 2017 to 2024, a total of 84 patients with audiovestibular manifestations have been diagnosed at our center with third-window syndromes. Within this cohort, we found that 92.86% (n = 78) corresponded to superior semicircular canal dehiscences, while the remaining six were attributed to otic capsule dehiscences at different levels. Thus, the remaining 7% of cases are intended to be described as follows.

*3.1. Case 1*

A 73-year-old male wearing hearing aids in both ears has bilateral hearing loss of five years of evolution, ear fullness, and tinnitus, more pronounced in the left ear, particularly in noisy environments. His hearing loss fluctuated without any vestibular symptoms. Otoscopic and otoneurological examinations were normal. Pure-tone audiometry in the free field revealed moderate hearing loss in the right ear (44 dB) and severe-to-profound loss in the left (up to 100 dB) (Figure 1A). An MRI showed normal retrocochlear findings. Speech audiometry indicated discrimination difficulties in the left ear (70 dB threshold for 50% speech intelligibility). Treatment involved a cochlear implant in the left ear and a hearing aid in the right. Complementary tests for vestibular function were normal. Typically, we do not perform CT scans of the ear petrosal bone, which we usually implant as part of our protocol, although we do perform cerebral MRIs to ensure normal cochlear morphology, as well as the internal auditory canal and facial canal. Additionally, in this case, the imaging test had already been requested even prior to the decision for implantation due to asymmetrical hearing loss.

During cochlear implant surgery, a 10-minute gusher phenomenon occurred. A CT scan revealed a lack of continuity between the cochlea and the internal auditory canal without facial nerve involvement (Figure 1B,C). However, word discrimination post-implantation was favorable after 5-months of follow-up.

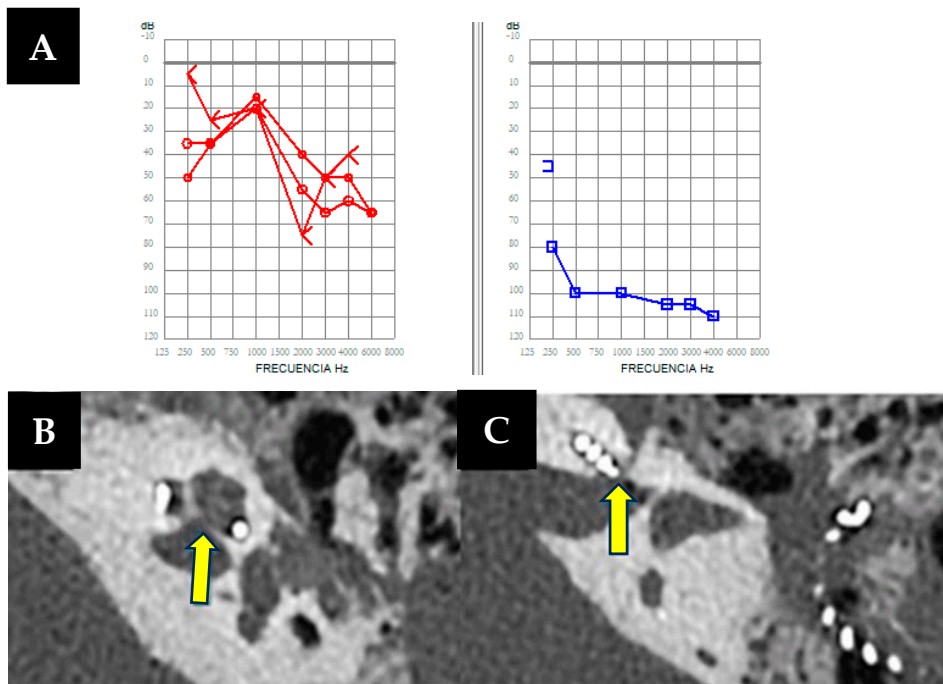

**Figure 1. (A)**. Free-field unilateral pure-tone audiometry showing a moderate hearing loss in the right ear (in red, 44 dB) and a severe to profound hearing loss in the left ear (in blue 100 dB). **(B,C)** Left axial petrous bone CT scan showing an air-fluid level in the mastoid cavity, as well as a bone defect (arrows) with dehiscence between the basal turn of the cochlea and the internal auditory canal.

*3.2. Case 2*

A 53-year-old man presented with bilateral hearing loss of two years duration, worsening in the right ear following a traumatic brain injury to the right temporoparietal region. He was a hearing aid user, reporting partial improvement, more pronounced in the left ear. He also experienced bilateral fullness and recurrent vertigo episodes lasting half a day daily, associated with Tumarkin's otolithic crises and concurrent hearing fluctuations. The otoscopic and neuro-otological assessments were unremarkable. Audiological tests as free field audiometry revealed severe hearing loss in the right ear (up to 80 dB) and moderate loss in the left (55 dB) (Figure 2A), predominantly affecting low frequencies. Vestibular tests showed decreased oVEMPs and hypofunction in specific semicircular canals, with vHIT exhibiting slightly reduced VOR gain values for both HSC and PSC on the right side (0.67 and 0.55, respectively), with mainly covert saccades along with refixation saccades. Clinically diagnosed Ménière's disease was confirmed, furthermore, through imaging findings with cerebral MRI and sequences of endolymphatic hydrops. Cochlear implant surgery in the right ear and intratympanic gentamicin in the left were indicated, despite contraindications for being bilateral (Webster et al. 2023 [11]). However, this patient experienced an average of up to two vertigo attacks per day, along with Tumarkin's crisis falls and a history of refractoriness to rescue treatments, including vestibular sedation, oral corticosteroid treatment, and even intratympanic dexamethasone instillations. Therefore, treatment options were very limited.

As an imaging test prior to implantation, the cerebral MRI previously requested when diagnosing Ménière's disease was used, which did not reveal any relevant findings at that time; hence, a pre-surgery cerebral CT scan was not requested. However, during cochlear implant surgery, a *gusher* phenomenon occurred [12], as shown in Video S1, followed by a postoperative CT revealing discontinuity affecting the cochlea and internal auditory canal, and the internal auditory canal and facial canal on the right side, without signs of temporal bone fracture. (Figure 2B,C). Despite treatment challenges, vertigo attacks were effectively managed, emphasizing the importance of individualized approaches in complex cases.

The finding in the computed tomography of the petrous bone of the implanted ear is relevant because, in relation to case number 1, it can be concluded that dehiscences between the cochlea and the internal auditory canal may justify both the *gusher* phenomenon observed during surgery and the clinical presentation with symptoms similar to Ménière's disease.

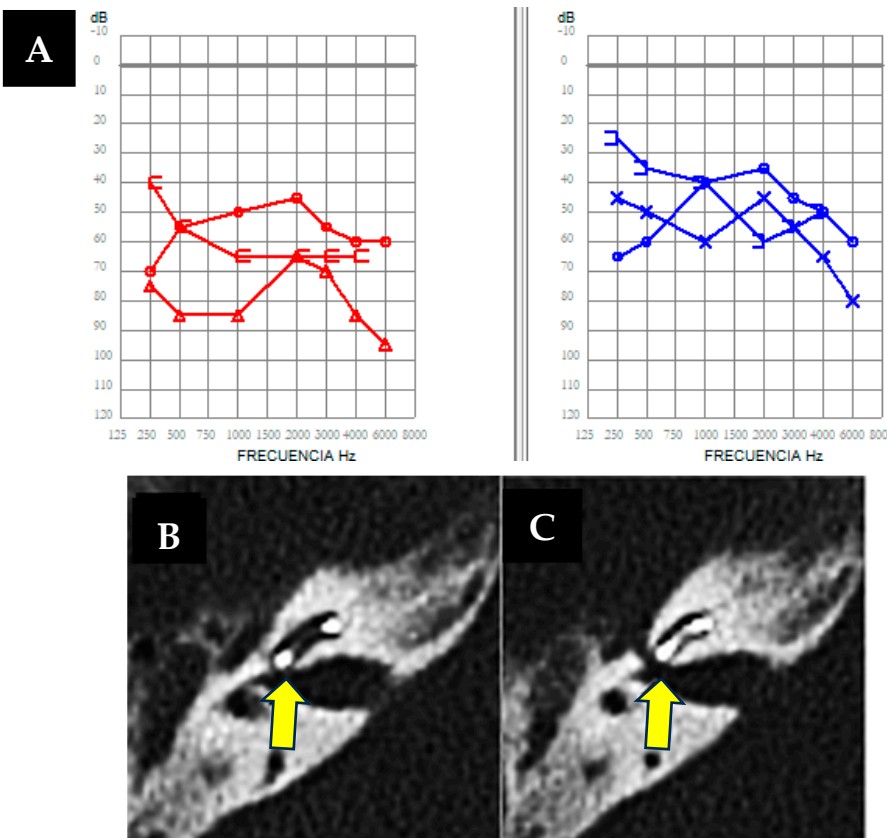

**Figure 2.** (**A**). Free-field bilateral audiometry was performed as the patient was wearing hearing aids, showing severe hearing loss in the right ear (in red) and moderate hearing loss in the left ear (in blue). (**B**,**C**) Axial right petrous bone CT scan showing a fluid-air level in the mastoidectomy cavity and a bone defect (arrows) with dehiscence between the basal turn of the cochlea and the internal auditory canal (IAC).

*3.3. Case 3*

A 54-year-old man presented with instability, multiple vertigo attacks, fluctuating hearing loss, tinnitus, and ear fullness in the right ear, with no clear triggers or history of autoimmune disease. Otoscopic examination revealed no anomalies, but neuro-otological evaluation showed persistent left nystagmus with head movements. Audiometry indicated mild hearing loss in the left ear (26 dB) and moderate sensorineural loss in the right (50 dB) (Figure 3A), with a retrocochlear pattern observed in speech audiometry for the right ear. VHIT with reduced VOR gain values for both HSC and PSC on the right side (0.52 and 0.65, respectively) with mainly covert saccades, and also decreased oVEMP tests confirmed a right vestibular deficit, and interaural asymmetry indices were consistent with vestibulopathy. A subsequent hydrops sequence revealed hyperintensity signals in the perilymph (as shown in Figure 3D,E) in the right basal cochlear turn in the real inversion recovery sequence, indicating a potential inflammatory component at that location. The hyperintensity signal extended to the adjacent portion of the facial nerve, allowing confirmation of the presence of a cochleo-facial dehiscence, as observed in the computed tomography of the petrous bone of the right ear (Figure 3C).

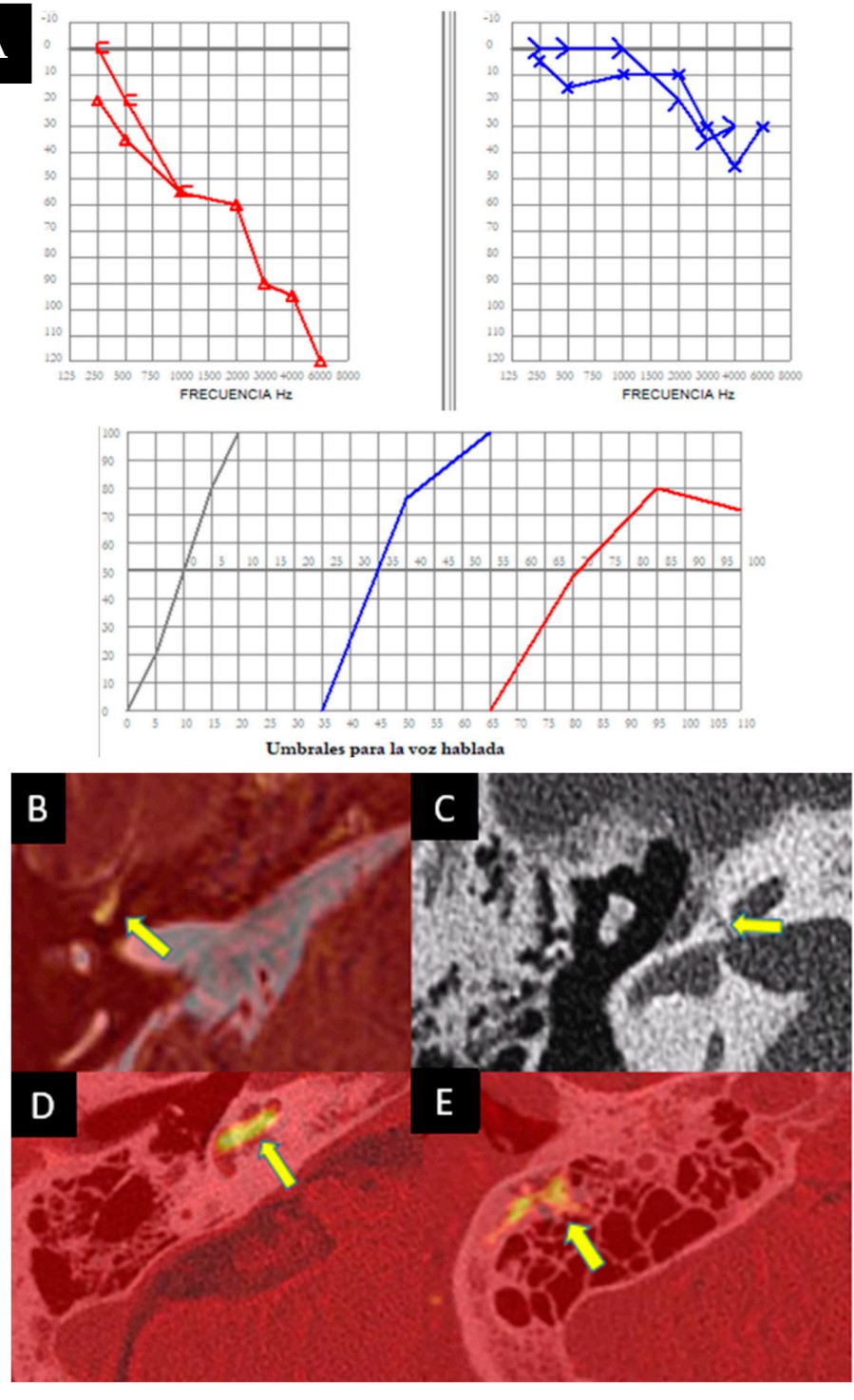

**Figure 3.** (**A**) Bilateral pure-tone audiometry showing mild hearing loss in the left ear (in blue) and moderate hearing loss in the right ear (in red), with speech audiometry indicating a retrocochlear pattern in the right ear. (**B**–**E**) MRI (Magnetic Resonance Imaging) in hydrops sequences showing marked hyperintensity of signal (arrow) in the perilymph in the right basal cochlear turn in the real inversion recovery sequence, which may indicate an inflammatory component. The extent of this signal hyperintensity necessitates ruling out a cochleofacial fistula component, which is confirmed (arrow) by performing an axial right petrous bone CT scan (**C**). (**D**,**E**) show axial fusion of cerebral MRI and CT showing potential inflammatory enhancement within the cochlea (arrow) and in the mastoid tip.

This is a very complex case. The initial diagnostic hypothesis is that there is a coexistence of Ménière's disease in its variant affecting high-frequency hearing, combined with a cochleo-facial fistula. Therefore, treatment with acetazolamide 250 mg/24 h was initiated, and the patient showed a favorable evolution.

### 3.4. Case 4

A 39-year-old woman presented with worsening instability lasting hours over a year, without vertigo attacks but with left ear tinnitus and progressive hearing loss persisting for over a decade. Otoscopic examination revealed left ear anomalies, including a monoaural anteroinferior area and type II atelectasis. Neuro-otological examination noted the right horizontal nystagmus synchronizing with cervical flexion and the right lateral head position. Pure tone audiometry indicated normal hearing in the right ear but moderate mixed hearing loss (52 dB) in the left, predominantly conductive (Figure 4A). A type B tympanogram suggested an otosclerosis suspicion. Vestibular exams showed normal operating gains and symmetrical VEMP responses. In this case, Ménière's disease is not suspected due to the absence of vertigo attacks, despite increased instability. Given the audiometric pattern with a significant difference between air and bone conduction, tympanometry, and the progressive and non-fluctuating nature of hearing loss, a computed tomography of the petrous bone of the left ear was requested to rule out otosclerosis before considering endolymphatic hydrops. It was confirmed that there were no signs of otosclerosis in the middle ear, but notably, it revealed cochlear-carotid dehiscence (Figure 4B,C). Ménière's disease was ruled out due to the absence of vertigo attacks. The patient was discharged with steroids at a dose of mg/kg when presenting auditory fluctuations, showing good control of the disease.

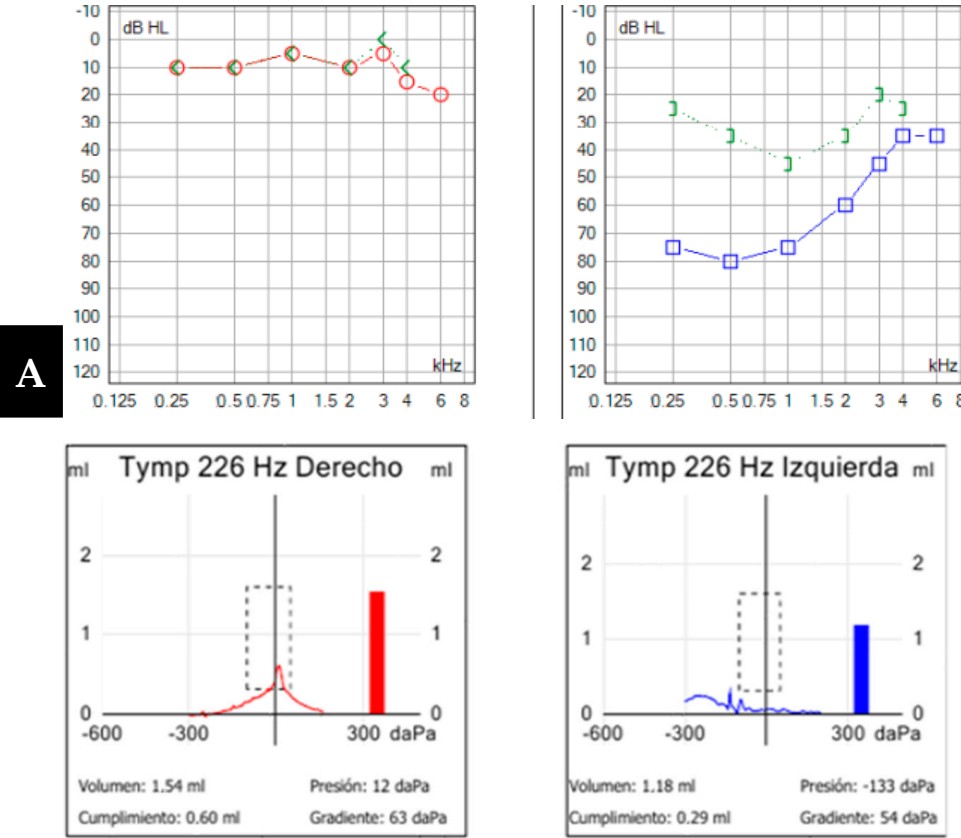

**Figure 4.** *Cont.*

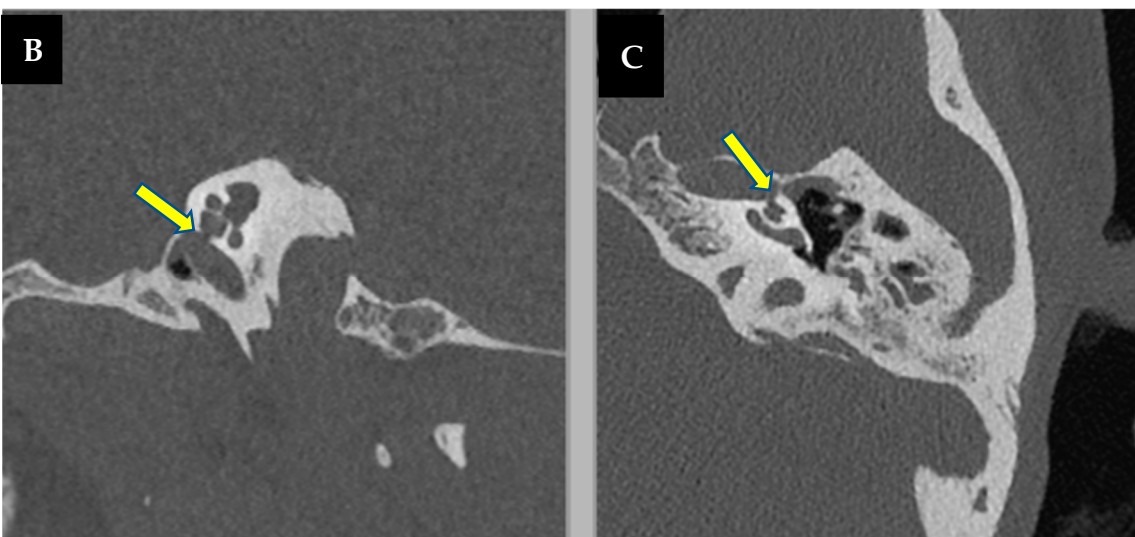

**Figure 4.** (**A**). Pure-tone audiometry showing normal hearing in the right ear (in red) and a moderate mixed-type hearing loss in the left ear (in blue), with a tympanogram classified as type B according to Jerger's classification. (**B,C**). Computed tomography image of the left petrous bone in the sagittal plane (**B**) and axial plane (**C**). Cochleo-carotid dehiscence is observed (yellow arrow) with a lack of bony coverage between the apical region of the cochlea and the carotid canal.

*3.5. Case 5*

An 18-year-old male, initially treated for a head injury and loss of consciousness, was presented with dizziness exacerbated by auditory stimuli and pressure changes, hearing loss, and a blockage sensation in the left ear that resolved after a few hours. Brain CT incidentally revealed left jugular bulb dehiscence into the ampullary region of the posterior semicircular canal, later confirmed on temporal bone CT (Figure 5A). Pure-tone audiometry during auditory fluctuation demonstrated conductive hearing loss, more pronounced in low frequencies (Figure 5B). Ménière's disease was ruled out due to the absence of vertigo attacks. The patient exhibited left irritative horizontal nystagmus and downbeat nystagmus during head hyperextension. The vHIT results were normal, while VEMPs showed decreased interaural asymmetry, predominantly left-sided. The diagnosis of left jugulo-vestibular dehiscence as third-window syndrome was made. Due to the repetition of his symptoms, the patient's torpid response to treatment and episodic worsening of dizziness prompted further investigation. A brain MRI with hydrops sequence was performed and revealed moderate vestibular hydrops in the left ear with mild cochlear involvement (Figure 5C,D). The patient was discharged with acetazolamide 125 mg/24 h with a decrease in symptoms during follow-up.

*3.6. Case 6*

A 46-year-old male presented with a six-week history of instability, auditory fluctuations, and ear fullness in the left ear, without vertigo attacks but significantly impairing him. Symptoms worsened with headphone use and the Valsalva maneuver. Otoscopic and neurotological examinations were normal. Complementary tests revealed mild hearing loss with a conductive component in the left ear (Figure 6A), alongside an absent stapedial reflex. vHIT showed no abnormalities, but ocular VEMPs indicated utricular hypofunction in the left ear.

As the patient did not meet the criteria for Ménière's disease and with the suspicion of otosclerosis and possible retrofenestral involvement, a temporal bone CT scan was requested. The scan revealed a dehiscence at the level of the jugular diverticulum with the vestibular aqueduct of the left endolymphatic sac (Figure 6B), along with a dehiscence also in that ear at the level of the superior semicircular canal (Figure 6C). Initially, the

patient rejected the introduction of medical treatment, although the persistence of his symptoms, along with two auditory fluctuations in three months, justified the performance of an MRI with hydrops sequences, which turned out positive for mild cochleovestibular endolymphatic hydrops. Subsequently, treatment with acetazolamide 250 mg/24 h was initiated, leading to good symptom control.

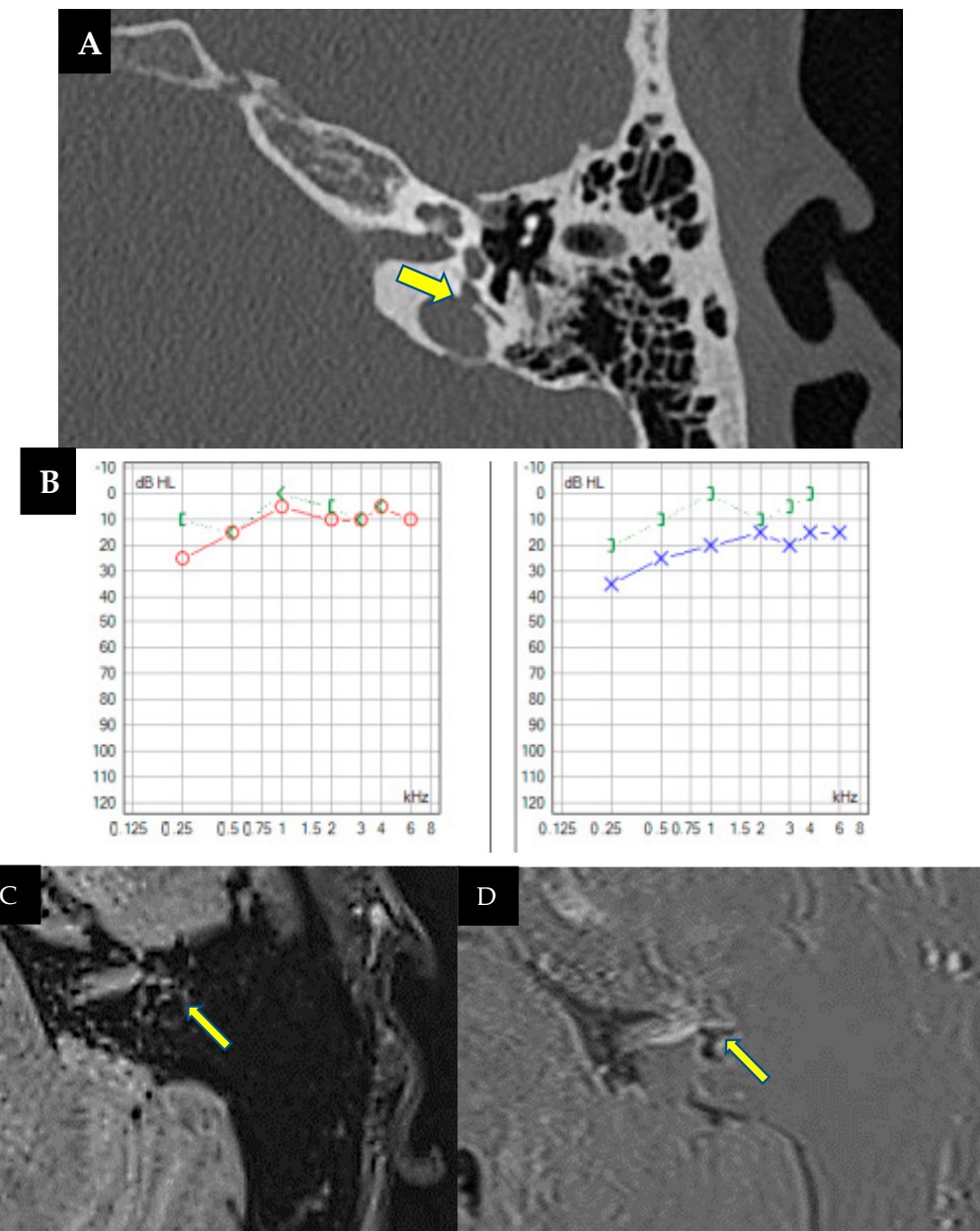

**Figure 5. (A)**. Computed tomography of the left petrous bone where the left ear shows a dominant and high jugular bulb with an apparent dehiscence towards the ampullary portion of the posterior semicircular canal (arrow). No dehiscence towards the middle ear is observed due to a thin bony membrane. (**B**). Pure tone audiometry showing normal hearing in the right ear (in red) and mild conductive hearing loss (26 dB) in the left ear (in blue), with significant impairment at low frequencies (250 Hz and 500 Hz). (**C,D**). MRI with T2 Flair 3D in endolymphatic hydrops sequences, axial view, focused on the membranous labyrinths. Dilatation of the saccule, confluent with the utricle, is observed, indicating moderate vestibular hydrops, possibly with a mildly associated cochlear component (**C**). Cerebral MRI with real inversion sequences and an axial view highlights and confirms the findings suggestive of the previously described sequence (**D**).

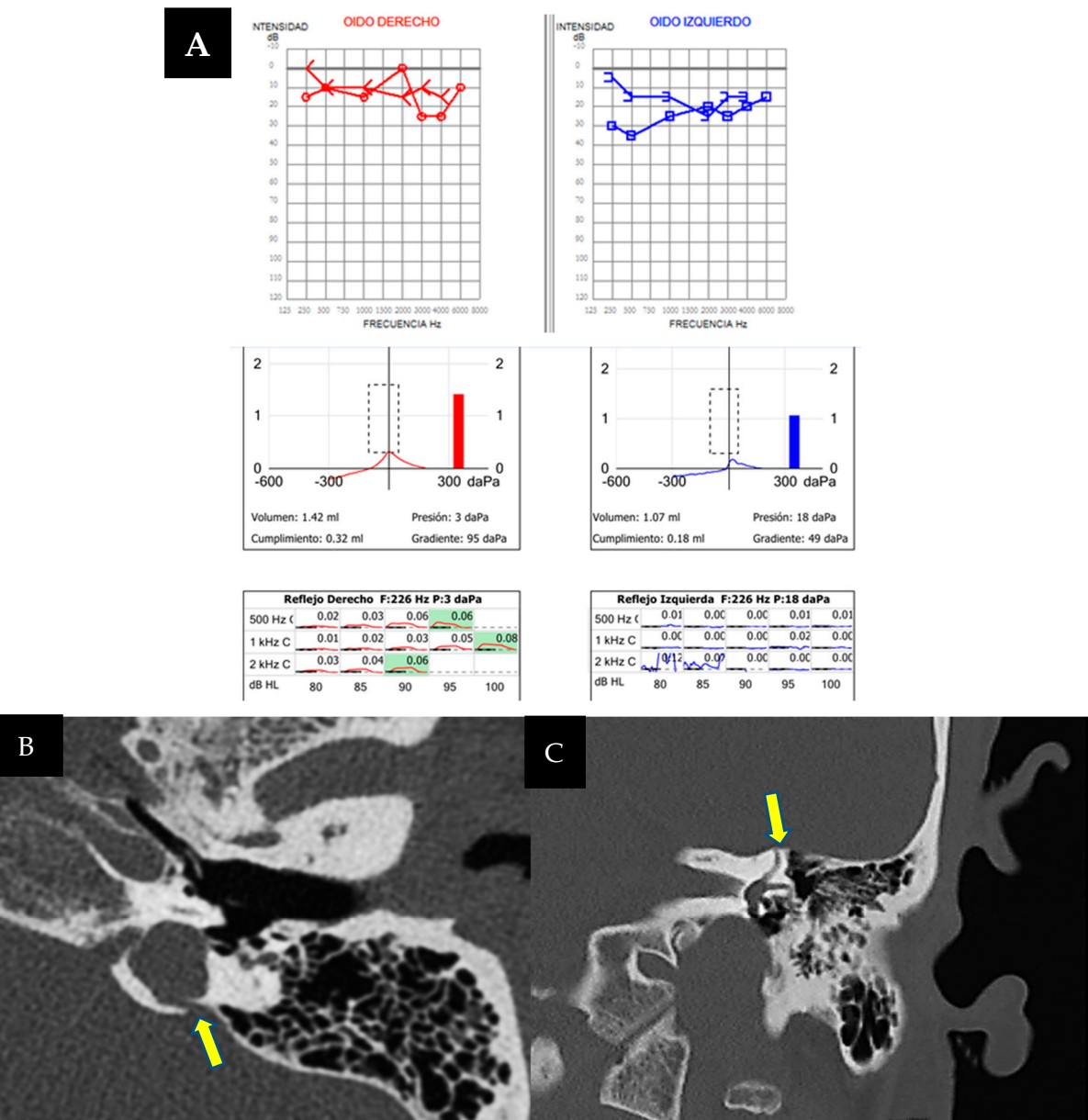

**Figure 6.** (**A**). Pure-tone audiometry showing mild conductive hearing loss in the left ear (in blue), with a Jerger type B tympanogram and the absence of a stapedial reflex on that side. (**B**,**C**) Left axial temporal bone CT showing jugular dehiscence with the vestibular aqueduct (yellow arrow). Coronal projection of the temporal bone CT on the left side, revealing a lack of continuity in the bony coverage of the superior semicircular canal (**C**).

In the study we present, it can be observed that a clinical presentation compatible with Ménière's disease or Ménière-like symptoms does not exclude the possibility of another condition resulting in a lack of continuity in the otic capsule. The demographic characteristics, main symptoms, and audiovestibular results, along with the final diagnosis, are summarized in Table 1.

**Table 1.** Summary of each clinical case. R: right ear; L: left ear; MD: Ménière disease; EH: endolymphatic hydrops; VA: vestibular acueduct; SSC: superior semicircular canal; PSC: posterior semicircular canal.

| | Pacient | Hipoacusis | Tinnitus/Fullness | Vertigo Attacks | Vestibular Tests | Other | md/eh | Final Diagnose |
|---|---|---|---|---|---|---|---|---|
| **CASE 1** | MAN 73 YEARS | R: 40 dB L:100 dB | YES | NO | VHIT NORMAL VEMP NORMAL | LEFT GUSHER | NO | DEHISCENCIE COCHLEO-ICA |
| **CASE 2** | MAN 53 YEARS | R: 80 dB L: 55 dB | YES | YES | VHIT ALTERED VEMP ALTERED | RIGHT GUSHER | YES | MD + DOUBLE DEHISCENCE |
| **CASE 3** | MAN 54 YEARS | R:50 dB L: 26 dB | YES | YES | VHIT ALTERED VEMP ALTERED | - | YES | DEHISCENCIE COCHLEA-FACIAL. + MD |
| **CASE 4** | WOMAN 39 YEARS | R:10 dB L: 52 dB | YES | NO | VHIT NORMAL VEMP NORMAL | - | NO | DEHISCENCIE COCHLEO-CAROTID |
| **CASE 5** | MAN 18 YEARS | R:7 dB L: 26 dB | YES | NO | VHIT NORMAL VEMP ALTERED | TULIO AND HENNEBERT | NO | DEHISCENCE JUGULAR-PSC |
| **CASE 6** | MAN 46 YEARS | R:12 dB L: 23 dB | YES | NO | VHIT NORMAL VEMP ALTERED | TULIO AND HENNEBERT | NO | DEHISCENCE YUGULAR-VA + DEHISCENCE SSC |

## 4. Discussion

The third mobile window syndrome has been a new and significant clinical diagnosis, proving to be less rare than initially believed. In 2016, Fang et al. [13] demonstrated in a study of 1060 temporal bones that the incidence of continuity alterations in the otic capsule, especially the cochlea-facial type (as in the first and third cases presented), was as high as 0.6%, with a higher risk among the Caucasian population. The recognition of new entities that can behave as a third-window phenomenon beyond the three listed in the differential diagnostic table of the Bárány Society consensus in 2015 [5] has not only increased the incidence but also expanded the range of symptoms they produce. These symptoms go beyond those typically related to otolithic dysfunction caused by sound and pressure (Tullio or Hennebert phenomenon) or autophony and may include cognitive disturbances, loss of spatial orientation, manifestations of anxiety, and migraine episodes. It is essential to consider that studies such as Chi et al. [14] in 2010 suggested that the size of the dehiscence might determine the presence or absence of symptoms. In their observation, fistulas larger than 2.5 mm tended to produce both audiovestibular symptoms, while those smaller than this size produced auditory or vestibular symptoms. This information must be taken into account to correlate clinical and radiological findings.

In 2019, Wackym PA et al. [15] conducted a literature review, calculating the approximate prevalence of optic capsule dehiscences gathered from a cohort of 401 patients with symptoms consistent with a mobile third window. They observed that, compared to the diagnoses in our patient cohort, cochleofacial fistula had a prevalence of 10.4%, followed by cochlea with carotid at 7.7%, cochlea with internal auditory canal at 1%, and finally posterior semicircular canal and jugular bulb at 0.2%. This fact not only reinforces the originality and extreme rarity of our findings but also asserts that in two of the cases,

both the second and sixth, where there is coexistent double dehiscence, had never before been described in the literature, not only as an incidental radiological finding but also as a generator of a mobile third-window syndrome.

Given this point, it is interesting to return to 2022, when Reynard et al. [16] published a proposal for an anatomoclinical and radiological classification for mobile third-window syndromes, aimed at improving diagnosis and appropriate therapeutic modalities. They created several subtypes, among which was type 1, with otic capsule (OC)-meningeal dehiscence; type II, which was OC-vascular (arterial or venous); and type III, or OC-petrosal, which included both the facial canal and internal auditory canals. Finally, there are two groups: one is called intralabyrinthine, the third mobile window-like variants or type IV and the other is called the multiple location (V) group. It can be deduced, as in our cohort, that both cases 1 and 3 would be type III, cases 4 and 5 would be type II as they involve vascular structures, and finally, both cases 2 and 6 would be in group V or multiple locations. Another finding from Reynard's article consistent with our work is the possibility to find between 14 and 40% of alterations in VEMPS in the form of reduced response, something that occurred in our case series in all those who presented alterations in this vestibular test, that is, in 4 out of 6.

Despite the lack of convergence regarding the prevalence and current status of otic capsule dehiscences as possible third-window phenomena, there does seem to be agreement that differences in the slice thickness and the equipment used in computed tomography are the most common causes of variations in the observed dehiscence frequency across different studies [15,17]. The prevalence of otic capsule dehiscence may differ among various patient populations. It has been observed that in ears with chronic otitis media, the prevalence of these entities is significantly higher compared to normal ears [18]. Similarly, Krombach et al. [19] observed in 2016 that, especially in patients with vertiginous or unstable clinical presentation, otic capsule dehiscences were diagnosed more easily than in cases with auditory symptoms or asymptomatic individuals.

In 2017, Gurkov et al. [20] began to report an increase in the presence of endolymphatic hydrops phenomenon secondary to abnormalities in the inner ear structure resulting from various lesions, as seen in 4 of our 6 patients, one of whom had definite Ménière's disease (MD) and the other an MD variant affecting high frequencies. Years ago, to try to quantify it, Naganawa et al. [21] observed a strong correlation between the presence of endolymphatic hydrops and semicircular canal dysplasia. Our work is among the first to report the relationship between other otic capsule dehiscences and the occurrence of hydrops in complementary tests, beyond superior semicircular canal dehiscence. Although the exact reason is not clearly defined, it is suggested that this association may be justified by the theory of changes in hydrostatic pressure at the level of the membranous labyrinth that can occur at the cochleovestibular level when there are bone dehiscences [22]. When a patient presents clinical symptoms compatible with otic capsule dehiscence, it is important to consider and establish a correct differential diagnosis with conditions such as Ménière's disease or other third-window syndrome, especially with perilymphatic fistula or vestibular aqueduct dilatation. As seen in the attached cases, especially in the first and third, the clinical picture and complementary tests suggested Ménière's disease.

Furthermore, what could explain the cerebrospinal fluid leakage during cochlear implant surgery in the first and second cases? The gusher phenomenon can serve as a sign of clinical presentation of otic capsule dehiscence in these cases, as, as proven, the underlying problem, which effectively existed in both patients, is a bony defect at the bottom of the internal auditory meatus. However, a better understanding and a deeper study of all these entities are still necessary, with an increase in the number of cases to be described to answer all these questions.

It is particularly interesting, as can be observed in the study, the variable pattern of involvement of these dehiscences in different patients. With this statement, we are not only referring to clinical manifestations but also to complementary tests. In other words, we have observed repercussions in all audiograms, in five out of six in vHIT and VEMPs,

in hydrops sequence MRI tests, and even in a couple more in a tympanogram that could simulate middle ear occupation and showed absence of the stapedial reflex. Therefore, we would like to emphasize the importance of conducting all tests that can provide relevant information in order to subsequently contribute to a better characterization of these entities.

The main limitation of this study is that, firstly, as a case series, it has a small sample size. However, it does not aim to conduct a defined statistical study, as it would be difficult to establish statistically significant relationships with such a small number of patients, although it does aim to provide a descriptive overview of these entities. The other limitation lies in those patients who were not undergoing surgery and were evaluated with medical treatment, as evidently the outcome of therapeutic management will be much less beneficial compared to surgical intervention. Often, clinical follow-up and potential improvements will be influenced by the patient's own subjective perception, which may not always be accompanied by improvement in objective tests.

## 5. Conclusions

Otic capsule dehiscences are relatively new and unknown entities, diagnosed only in cases of clinical suspicion. It is necessary to consider the possibility of otic capsule dehiscence and third-window syndrome in cases that clinically suggest Ménière's disease, with discrepancies in complementary tests or a slow response to treatment. Although there are highly sensitive and specific tests, such as VEMPs, to suspect these entities, it is necessary to complete the study with imaging tests, especially computed tomography (CT) of the petrous bone, to locate and characterize the defect in the otic capsule responsible for the clinical picture.

Among other things, this work has the role of sensitizing ENT specialists to the fact that there are several variants of dehiscence of the otic capsule without SSCD. Many ENTs and radiologists only focused on finding SSCD and ignored the existence of other forms of OCD; thus, the diagnosis of this mimicking pathology is missed. Further research into otic capsule dehiscences is needed to describe the types of clinical manifestations they are responsible for. In our cases, for now, we cannot always justify the patients' clinical symptoms based on these findings.

**Supplementary Materials:** The following supporting information can be downloaded at: https://www.mdpi.com/article/10.3390/audiolres14020032/s1.

**Author Contributions:** Conceptualization, C.P.-M. and J.L.-P.; methodology, C.P.-M. and J.L.-P.; software, J.L.-P.; validation, C.P.-M. and J.L.-P. formal analysis P.D. and J.L.-P.; investigation, J.L.-P., C.P.-M. and R.M.-H.; resources C.P.-M.; data curation, J.L.-P.; writing—original draft preparation, J.L.-P.; writing—review and editing J.L.-P., O.G. and C.P.-M.; visualization, C.P.-M. and J.L.-P.; supervision, C.P.-M. and M.M.; project administration, R.M.-H.; funding acquisition, not required. All authors have read and agreed to the published version of the manuscript.

**Funding:** This research received no external funding. No third-party funding was received for this research.

**Institutional Review Board Statement:** This study protocol was reviewed in November 2023 and approved by Ethics committee of Clínica Universidad de Navarra, approval number (CEI 2023.163). Even more, this study was designed and performed in accordance with the ethical guidelines of the 1975 Declaration of Helsinki. Written informed consent was obtained from all to participate in the study. Even though, written informed consent was obtained from the patient for publication of the details of their medical case and any accompanying images.

**Informed Consent Statement:** Informed consent was obtained from all subjects involved in the study.

**Data Availability Statement:** Data pertaining to this study can be shared upon request to the corresponding author.

**Conflicts of Interest:** The authors report no conflicts of interest. The authors alone are responsible for the content and writing of the paper.

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
