# Peer review of "Otic Capsule Dehiscences Simulating Other Inner Ear Diseases: Characterization, Clinical Profile, and Follow-Up—Is Ménière’s Disease the Sole Cause of Vertigo and Fluctuating Hearing Loss?"

_audiolres, doi:10.3390/audiolres14020032_

Round 1

Reviewer 1 Report

Comments and Suggestions for Authors

I enjoy readind the manuscript and  I think the topic is original and for this reason suitable for publication.

Only a few consideration.

- I think the introduction section can focus better to the aim of the manuscript while maybe some specific consdieration about the explanation of the origin of the capsule dehiscences can be reduced.

- In material and methods section I think that it is not important to specify how cphlear implant surgery was performed. It is sufficient to write, as done in line 108, that the procedure was standard. 

- Could the authors justify why in case 1 and 2 the basal audiogram was free field, please?

- Coulfd the authors justify why before CI surgery a CT scan was not performed please?

Comments on the Quality of English Language

I think the manuscript is well comprehensible but some sentences are not well written and I minor english revision could improve the quality of the work

Author Response

 I think the introduction section can focus better to the aim of the manuscript while maybe some specific consdieration about the explanation of the origin of the capsule dehiscences can be reduced--> Thank you very much for your comment. You will see that we have simplified the section regarding the embryogenesis of the otocyst, keeping only the embryological reference to avoid redundancy.

-In material and methods section I think that it is not important to specify how cphlear implant surgery was performed. It is sufficient to write, as done in line 108, that the procedure was standard--> Thank you for your input, and we agree with your suggestions. We have simply stated that the standard procedure was performed, as you will see in line 110.

Could the authors justify why in case 1 and 2 the basal audiogram was free field, please?--> We apologize as it has been our mistake. I have explained that both patients had previously been diagnosed with progressive bilateral hearing loss at another center, necessitating hearing aids. That is why we carried out a free-field audiometry with hearing aids, as you will read regarding case 1, in lines 125 and 128. On the other hand, regarding case 2, we have justified why it is indicated to perform a free-field audiometry, both in the caption of figure 2A and in the case description, in line 151, where we explain the reason why the patient was wearing hearing aids and the sensation of asymmetric performance he experienced with them. This fact also prompted us to conduct the free-field audiometry to observe if there was indeed an unequal gain.

Coulfd the authors justify why before CI surgery a CT scan was not performed please?-->Thank you for your comment, which I will respond to from two perspectives. Initially, there are clinical guidelines that advocate for MRI as a good alternative as a pre-intervention test, and we prefer to carry it out ahead of CT scans. We have encountered, on multiple occasions, patients with central level tumors, which not only could contraindicate surgery at an initial stage but could also cloud the prognosis even with the implant. On the other hand, and this has been added to the manuscript after your review, concerning the cases in our article, in the first patient, we had previously performed a cerebral MRI to rule out retrocochlear pathology due to the marked audiometric asymmetry (lines 130 and 137). In the second patient, we also used a cerebral magnetic resonance imaging as a pre-surgery test that we had previously requested after diagnosing Ménière's disease (which, as per Bárany's criteria, is clinical), as we routinely do. This is not only to rule out other concomitant central diseases but also because as a protocol, in patients with Ménière's disease, we also request cerebral MRI and hydrops sequences to assess the degree of cochleo-vestibular involvement. We explained this in line 170. I hope I have explained myself well, both here and in the manuscript.

I think the manuscript is well comprehensible but some sentences are not well written and I minor english revision could improve the quality of the work--> Thank you for your latest comment. We have reviewed the manuscript again and modified long sentences, attempting to simplify them to ensure the message is clear and concise, avoiding conjunctions that might complicate the reading. Thank you very much for your dedication.

Reviewer 2 Report

Comments and Suggestions for Authors

I consider this literature to be well-described, focusing on an intriguing subject as a case series. Considering that temporal bone CT imaging is not a particularly challenging procedure, it might be worth considering the possibility of otic capsule dehiscence in patients with vertigo symptoms and a slight A-B gap.

I don't have any specific requests, but I would appreciate it if Figure 2A could be reconfigured. Alternatively, adding some more detailed explanations would be helpful.

Comments on the Quality of English Language

Nothing to add.

Author Response

I consider this literature to be well-described, focusing on an intriguing subject as a case series. Considering that temporal bone CT imaging is not a particularly challenging procedure, it might be worth considering the possibility of otic capsule dehiscence in patients with vertigo symptoms and a slight A-B gap-> Thank you very much for your contribution, which we found very interesting and indeed is one of the key messages of our work. Furthermore, we have included it both in the abstract (line 15) at the end of the introduction, as well as in line 63.

I don't have any specific requests, but I would appreciate it if Figure 2A could be reconfigured. Alternatively, adding some more detailed explanations would be helpful. -> Thank you again for your comment. As you suggested, we have justified why it is indicated to perform a free-field audiometry, both in the caption as requested for reconfiguration, and in the case description, in line 151, where we explain the reason why the patient was wearing hearing aids and the sensation of asymmetric performance he experienced with them. This fact also prompted us to conduct the free-field audiometry to observe if there was indeed an unequal gain.